# Method and Device for Measuring the Diameter at Breast Height and Location of Trees in Sample Plots

Shangyang Li [1,2], Xiaorong Chen [3], Yuanjing Sun [1,2], Chaihui Lv [4], Fangxing Yuan [1,2] and Luming Fang [1,2,*]

[1]  Key Laboratory of Forestry Intelligent Monitoring and Information Technology Research of Zhejiang Province, Zhejiang A & F University, Hangzhou 311300, China; lishangyang@stu.zafu.edu.cn (S.L.); 2020611011040@stu.zafu.edu.cn (Y.S.); 2019611011037@stu.zafu.edu.cn (F.Y.)

[2]  College of Mathematics and Computer Science, Zhejiang A & F University, Hangzhou 311300, China

[3]  Qingyuan Conservation Center of Qianjiangyuan-Baishanzu National Park, Lishui 323800, China; qybszcxr@163.com

[4]  Hangzhou Ganzhi Technology Co., Ltd., Hangzhou 311300, China; panjk@zafu.edu.cn

*  Correspondence: fluming@126.com; Tel.: +86-18968156768

**Abstract:** The diameter at breast height (DBH) and location of trees are important factors when surveying forest resources and ecological functions. In this study, a device mainly comprising a self-made DBH-measuring instrument and positioning base station was used. The hardware consisted of two devices to simultaneously measure the DBH and location of trees within a sample plot. Specifically, DBH is acquired by processing angle data with an algorithm, and locations are obtained by a five-sided ranging and positioning algorithm based on the received signal strength indicator and ultra-wideband (UWB) sensor. Data uploading, storage and analysis are performed by an upper computer. The device was used for the actual measurement of eight 24 m × 24 m square plots. The measurements of this device are essentially consistent with those of the DBH tape and calliper, with biases of −0.03 cm and −0.29 cm, respectively. Compared with a compass for location measurement, the device had a mean range bias of 25.41 cm, overall bias along the *X*-axis of 2.40 cm and overall bias along the *Y*-axis of 1.99 cm. Therefore, the device is considered to be sufficiently portable and practical and can reduce the heavy workload for surveyors to meet the requirements of accurate and smart measurements in forest resource surveys.

**Keywords:** sample plot factor; angle sensor; ultra-wideband sensor; received signal strength indicator algorithm

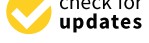



## 1. Introduction

Direct measurement and its derived factors are basic components of tree measurement which are of great significance for forest operation, management, planning and protection [1,2]. Specifically, diameter at breast height (DBH) reflects the annual rings, height and crown size of trees in surveys of forest stock volume and biomass [3]. Traditional manual methods of measuring DBH rely on tools such as DBH tape or calliper, which have several limitations, including low measurement efficiency due to manual reading and recording [4,5]. In recent years, several studies have further explored DBH measurement methods with varying success rates, such as ground-based light detection and ranging [6,7], close-range photography (CRP) [8,9] and smartphones with time-of-flight (TOF) cameras [10,11]. However, these DBH extraction methods based on point cloud computation have the drawbacks of complicated data processing, high cost and poor adaptability, making them unsuitable for wide application in forestry [12]. In recent years, my team has devoted itself to the research and development of forest survey tools and introduced a fast DBH measurement device that quickly measures DBH size and tree height. However, the device was severely affected by the weather, such as strong sunlight, and did not function properly [13]. A year ago, our team proposed a

new method for DBH measurement using a self-resetting displacement sensor; however, this method is prone to large errors when measuring trees with a DBH larger than 60.6 cm [14].

In addition to estimating DBH, acquiring the location of trees has great ecological significance for predicting tree growth and trends in species development [15]. Traditionally, tree location data are mainly acquired by close traversal using a compass and 100 m tape. However, this is a labour-intensive method [16,17] in which tree locations are acquired by computing the manually measured survey data. Various positioning measurement methods have emerged, including global navigation satellite systems [18], CRP [8,9], TOF cameras [10,11], etc., but these methods are easily affected by the surrounding environment and take a long time [19]. Therefore, a set of integrated devices that can rapidly and accurately measure tree DBH and location within sample plots is urgently needed. With the advancement of information technology, the development of these integrated devices is possible. Haglöf Company launched a similar device, but it was expensive [20]. Several years ago, we began to experiment with the integration of multiple tree factors with larger, more complex device designs [21]. This time, we have improved the equipment while retaining the better parts of the previous equipment, and developed a new positioning algorithm. The Hall-effect angle-position sensor is an angle-extraction sensor based on the changes in magnetic fields and has been widely used owing to its extremely high adaptability to different environments [22–24]. Ultra-wideband is a communication technology that uses discrete pulses with a duration of less than a nanosecond and can be applied in ranging and location measurements owing to its high penetration [25–27]. The environment of forest surveys is complex and the demands are high, and the angle sensor and UWB are low-cost, have stable performance and meet the needs of forest surveys [28,29].

In this study, a device integrating the above technologies to achieve factor measurements within a forest sample plot, including DBH and location, is proposed. Compared with the original equipment, we use a new structure and algorithm to improve the stability and anti-interference ability of the device and realise the measurement of multiple tree factors. The stability, high efficiency and feasibility of the equipment were verified by field tests using different environmental samples. The device can effectively reduce the tedious measurement work performed by surveyors in order to meet the higher requirements of precise and intelligent measurements for forest resource surveys.

## 2. Material and Methods

### 2.1. Mechanical Structural Design

The equipment was designed primarily based on actual forestry surveys and several trips to forest farms for practical graduate examinations, incorporating the unique elements of the forest. We consider that the use of equipment in the forest requires that it be waterproof and stable when installed. The device mainly comprises a self-made DBH-measuring instrument and positioning base station. The former component integrates DBH measurement and label-positioning functions. The main components are shown in Figure 1. The electrical circuit part of the DBH-measuring instrument consists of a microprocessor (STC, Shanghai, China), key (Dongke, Beijing, China), power supply (Zhongsun, China), Hall-effect angle-position sensor (Xingyang, China), compass (Jiage, China), UWB module (Lianwang, China), organic light-emitting diode (OLED) module (Sanluyi, China), secure digital (SD) card (Eansdi, Shenzhen, China) and Bluetooth (HC, Hongkong, China). The electrical circuit part of the base station consists of a microprocessor, power supply, gyroscope, compass, UWB module and OLED module. The main parameters of the device are shown in Table 1. The above design takes into account not only the accuracy of the measurement data of the device, but also the particularities of the forest terrain [21].

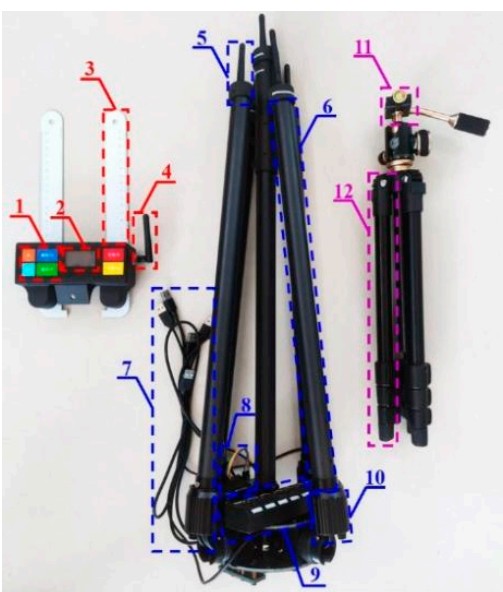

**Figure 1.** Structure diagram of the main components: 1: membrane key; 2: display; 3: arm ruler; 4: UWB label antenna; 5: UWB base station antenna; 6: carbon fibre tube; 7: power supply cable; 8: base station motherboard; 9: power supply port; 10: fixing knob; 11: fixing button; 12: supporting leg.

**Table 1.** Module parameters.

| Component | Chip Interface/Type | Interface Type | Quantity | Parameter | Function |
|---|---|---|---|---|---|
| Microprocessor | STC15W4K56S4 | IO port, SPI, etc. | 2 | SRAM: 4 KB; Flash: 56 KB | Data processing |
| Power-supply-management circuit | TP4056, AMS1117, etc. | DC voltage | 2 | Input: 3.7 V–5 V; Output: 5 V, 3.3 V | Power-supply-management |
| battery | Lithium–ion battery | DC 5.5–2.1 | 2 | 4000 mAh | Power supply |
| Gyroscope module | JY901B | Serial port | 1 | Measurement accuracy | Attitude measurement |
| Compass module | GY-26 | Serial port | 2 | Measurement accuracy: 0.1° | Azimuth angle measurement |
| UWB module | D-DWM-PG1.7 | Serial port | 6 | Measurement accuracy: 1 cm Measurement range: 0–150 m | Distance measurement |
| Hall-effect angle-position sensor | TM003A | Serial port | 1 | Measurement accuracy: 0.01 mm | Angle measurement |
| Analogue–digital converting module | ADS1115 | I2C | 1 | 16-bit, 4-channel | Analogue–digital conversion |
| Membrane key | PVC | Digital signal | 1 | 6 keys | Data recording |
| Display screen | OLED | SPI | 2 | 128 × 64 pixels | Data display |
| Bluetooth | HC-06 | Serial port | 1 | Communication distance | Upper computer communication |
| SD card | MicroSD | SPI | 1 | 2 GB | Data storage |

*2.2. System Process*

In setting up the system, it is necessary to start from an actual forest survey and combine the features of the forest factors to enable the integration of measured, uploaded and analysed data. The system process is shown in Figure 2; from bottom to top, the object, physical, hardware and data and software layers are presented. The sub-programmes of the software include the human–computer interaction programme for the keyboard input and

display control. The software layer includes a human–computer interaction programme for the button input and display control. The sampling programme is focused on sampling DBH and location information, and the data management programme is for data storage and communication. After completing the sample plot factor measurement, the DBH data are automatically consolidated into a series of comprehensive data to be uploaded to the data management system to fulfil the integrated measurement process [21].

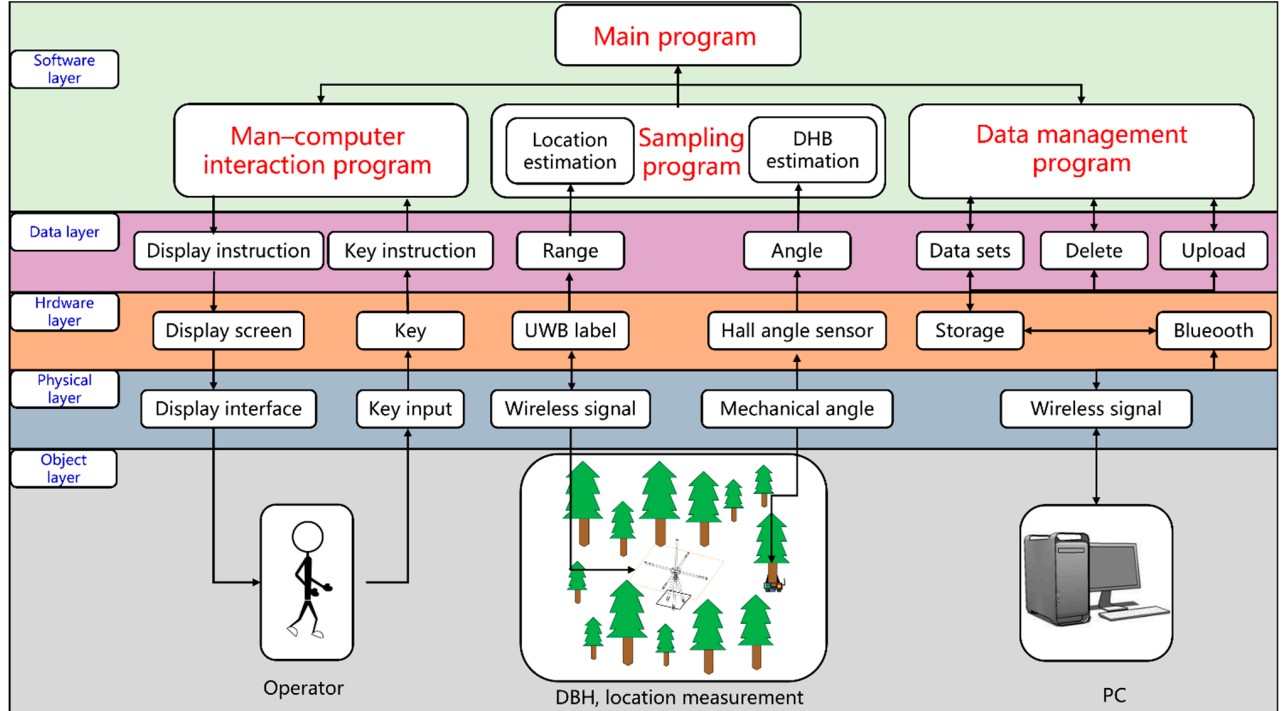

**Figure 2.** System flowchart.

### 2.3. DBH Measurement Algorithm

DBH measurement is shown in Figure 3, where A, B and C are contact points between the device and the tree trunk and two arcs BA and BC are generated. As tree trunks have irregular round shapes, the centres of two circles, $O_1$ and $O_2$, and two radii, $r_1$ and $r_2$, are generated.

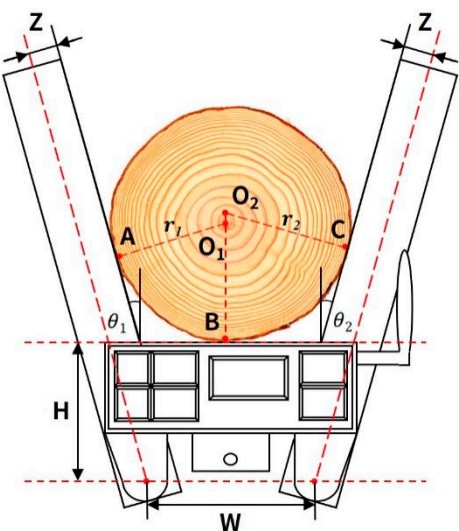

**Figure 3.** Schematic of diameter at breast height measurement.

For the DBH measurement instrument, Z (1.5 cm) is half the width of the arm ruler, H (6.85 cm) is the vertical distance between the top of the control box and centre of the rotation of the Hall-effect angle sensor, and W (9 cm) is the distance between the centres of rotation of the two Hall-effect angle sensors. To achieve rapid and accurate estimation of low-performance single-chip microcomputers, the Taylor expansion is used [30], and the value of $r_i$ ($i$ = 1, 2) can be obtained with Equation (1):

$$r_i = \frac{W \times \left(1 - \frac{\theta_i^2}{2!} + \frac{\theta_i^4}{4!} - \frac{\theta_i^6}{6!}\right) + 2 \times H \times \left(\theta_i - \frac{\theta_i^3}{3!} + \frac{\theta_i^5}{5!} - \frac{\theta_i^7}{7!}\right) - Z}{2 \times \left(1 - \theta_i + \frac{\theta_i^3}{3!} - \frac{\theta_i^5}{5!} + \frac{\theta_i^7}{7!}\right)} \qquad 0° < \theta_i < 90° \quad (1)$$

DBH, denoted by $d$, can be obtained with Equation (2):

$$d = r1 + r2 \tag{2}$$

### 2.4. Tree-Position Measurement Algorithm

The horizontal planes of the base stations are the coordinates of the X- and Y-axes, where $\vec{OE}$ is the positive direction of the X-axis and $\vec{OB}$ is the positive direction of the Y-axis, $\vec{OA}$ is the positive direction of the Z-axis and $|OA| = |OB| = |OC| = |OD| = |OE| = 0.5$ m. When measuring location, the distance between positioning label Q and each base station can be derived according to the 2-sided and 2-way ranging principle, as shown in Figure 4.

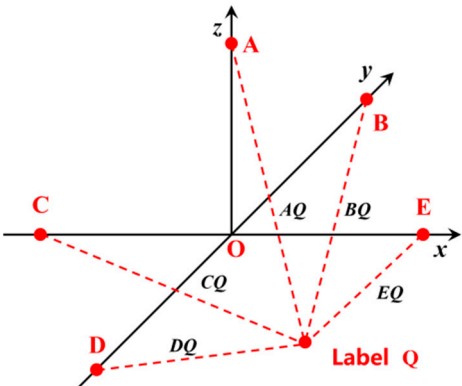

**Figure 4.** Schematic of position measurement.

The coordinates of base stations A–E and positioning label Q were set as follows: A (XA, YA, ZA), B (XB, YB, ZB), C (XC, YC, ZC), D (XD, YD, ZD), E (XE, YE, ZE) and Q (XQ, YQ, ZQ). The distance between the positioning label and each base station can be obtained with the following equations:

$$AQ^2 = \left(X_A - X_Q\right)^2 + \left(Y_A - Y_Q\right)^2 + \left(Z_A - Z_Q\right)^2 \tag{3}$$

$$BQ^2 = \left(X_B - X_Q\right)^2 + \left(Y_B - Y_Q\right)^2 + \left(Z_B - Z_Q\right)^2 \tag{4}$$

$$CQ^2 = \left(X_C - X_Q\right)^2 + \left(Y_C - Y_Q\right)^2 + \left(Z_C - Z_Q\right)^2 \tag{5}$$

$$DQ^2 = \left(X_D - X_Q\right)^2 + \left(Y_D - Y_Q\right)^2 + \left(Z_D - Z_Q\right)^2 \tag{6}$$

$$EQ^2 = \left(X_E - X_Q\right)^2 + \left(Y_E - Y_Q\right)^2 + \left(Z_E - Z_Q\right)^2 \tag{7}$$

Using $3 \times 3$ combinations of Equations (3)–(7), 10 groups of coordinates of the positioning labels can be obtained by solving the ternary equations: Q1 (X1, Y1, Z1), Q2 (X2, Y2, Z2), Q3 (X3, Y3, Z3), Q4 (X4, Y4, Z4), Q5 (X5, Y5, Z5), Q6 (X6, Y6, Z6), Q7 (X7, Y7, Z7), Q8 (X8, Y8, Z8), Q9 (X9, Y9, Z9) and Q10 (X10, Y10, Z10). Despite the high penetration capacity of the UWB wireless signal, it can be disturbed by the complicated forest environment, resulting in signal attenuation and prolonged communication time, and, consequently, excessively large values of AQ', BQ', CQ', DQ' and EQ' and their failure to intersect, as shown in Figure 5.

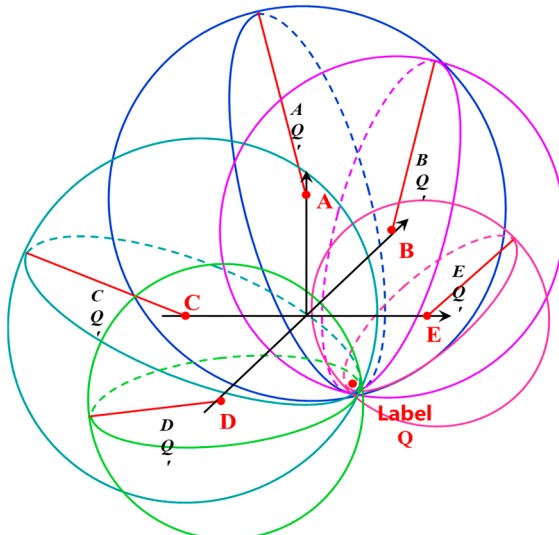

**Figure 5.** Schematic of actual position measurement.

To effectively address the problem, the weighted received signal-strength indicator algorithm can be used. As tree positioning is conducted using coordinates on a two-dimensional plane, the projected coordinates of Q on the XOY plane are needed to position the trees in the sample plot:

$$
\begin{cases}
XQ = \dfrac{\frac{X1}{AQ+BQ+CQ} + \frac{X2}{AQ+BQ+DQ} + \frac{X3}{AQ+BQ+EQ} + \frac{X4}{AQ+CQ+DQ} + \frac{X5}{AQ+CQ+EQ} + \frac{X6}{AQ+DQ+EQ} + \frac{X7}{BQ+CQ+DQ} + \frac{X8}{BQ+CQ+EQ} + \frac{X9}{BQ+DQ+EQ} + \frac{X10}{CQ+DQ+EQ}}{\frac{1}{AQ+BQ+CQ} + \frac{1}{AQ+BQ+DQ} + \frac{1}{AQ+BQ+EQ} + \frac{1}{AQ+CQ+DQ} + \frac{1}{AQ+CQ+EQ} + \frac{1}{AQ+DQ+EQ} + \frac{1}{BQ+CQ+DQ} + \frac{1}{BQ+CQ+EQ} + \frac{1}{BQ+DQ+EQ} + \frac{1}{CQ+DQ+EQ}} \\[2em]
YQ = \dfrac{\frac{Y1}{AQ+BQ+CQ} + \frac{Y2}{AQ+BQ+DQ} + \frac{Y3}{AQ+BQ+EQ} + \frac{Y4}{AQ+CQ+DQ} + \frac{Y5}{AQ+CQ+EQ} + \frac{Y6}{AQ+DQ+EQ} + \frac{Y7}{BQ+CQ+DQ} + \frac{Y8}{BQ+CQ+EQ} + \frac{Y9}{BQ+DQ+EQ} + \frac{Y10}{CQ+DQ+EQ}}{\frac{1}{AQ+BQ+CQ} + \frac{1}{AQ+BQ+DQ} + \frac{1}{AQ+BQ+EQ} + \frac{1}{AQ+CQ+DQ} + \frac{1}{AQ+CQ+EQ} + \frac{1}{AQ+DQ+EQ} + \frac{1}{BQ+CQ+DQ} + \frac{1}{BQ+CQ+EQ} + \frac{1}{BQ+DQ+EQ} + \frac{1}{CQ+DQ+EQ}}
\end{cases}
\tag{8}
$$

### 2.5. Experimental Process

We carried out field investigation using the equipment at a sample site. The measurement steps were as follows:

1. Data were collected from the gyroscope and compass modules on the base station, with base stations B, C, D and E on the same plane, and base station B was pointed due north at the correct location within the sample plot, as shown in Figure 6.
2. The sample plot number on the function interface of the DBH measurement instrument was set.
3. Based on the data of the compass module of the measurement instrument, the DBH of each tree within the sample plot was measured. Subsequently, the location information was acquired. (The DBH and location measurements are shown in Figure 7).

4. Upon completing the measurement, the data were uploaded to the upper computer for statistical analysis.

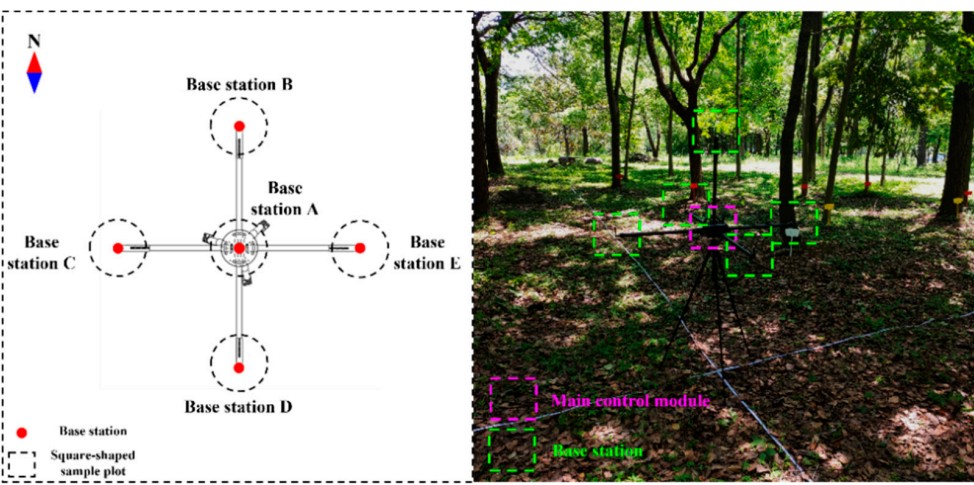

**Figure 6.** Base-station placement diagram.

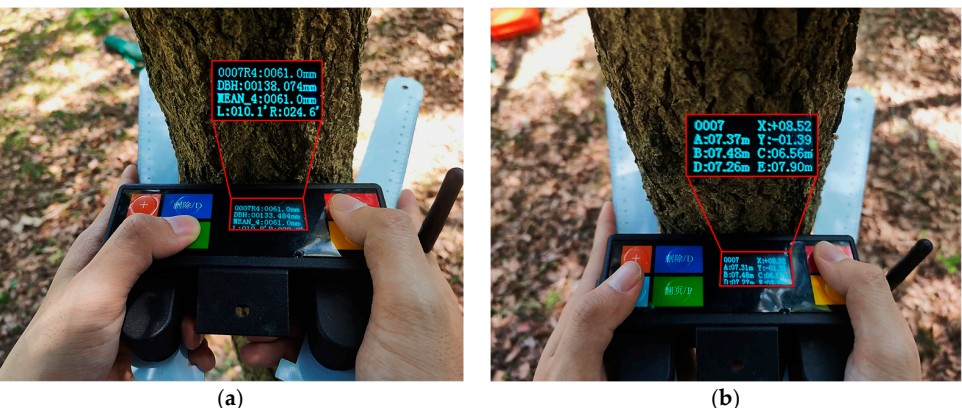

(**a**)                                     (**b**)

**Figure 7.** Measurement of (**a**) tree DBH and (**b**) tree location.

### 3. Results

*3.1. Experimental Location and Objects*

This experiment was conducted at the Donghu Campus of Zhejiang A & F University, located in the east of Hangzhou, Zhejiang Province (30°15′ N, 119°43′ E). Eight 24 m × 24 m square-shaped sample plots encompassing 200 trees were selected. The information of the eight sample plots is shown in Table 2. Specifically, sample plots 1–5 were mainly covered by fallen leaves, whereas plots 7 and 8 were covered by dense weeds.

**Table 2.** Sample information.

| Plot No. | Number of Trees | Species * | Slope (°) | DBH (cm) | | |
|---|---|---|---|---|---|---|
| | | | | **Min Value** | **Max Value** | **Mean** |
| 1 | 27 | S3, S4 | 3.2 | 6.01 | 35.65 | 22.25 |
| 2 | 28 | S1, S3, S4, S5, | 4.7 | 6.20 | 43.90 | 26.94 |
| 3 | 19 | S2 | 1.1 | 8.50 | 22.00 | 17.76 |
| 4 | 23 | S3, S5, S6 | 3.9 | 7.45 | 41.70 | 25.35 |
| 5 | 25 | S3, S4 | 5.3 | 7.5 | 37.48 | 21.59 |
| 6 | 27 | S1, S2, S3 | 12.5 | 9.08 | 38.88 | 22.83 |
| 7 | 26 | S3, S4, S5, | 15.6 | 7.00 | 39.72 | 19.60 |
| 8 | 25 | S1, S3, S4, S7 | 19.7 | 6.00 | 39.80 | 19.75 |

* S1: *Sapindus mukurossi*; S2: *Magnolia grandiflora*; S3: *Cinnamomum camphora*; S4: *Tulipa gesneriana*; S5: *Magnolia denudata*; S6: *Pinus massoniana*; S7: *Dalbergia hupeana*.

### 3.2. Similarity Estimation

In forestry, DBH data are typically measured using DBH tape and calliper. We thus compared the data measured by our device for each tree with the corresponding calliper and DBH tape values to estimate the availability of the device.

To validate the accuracy of the DBH estimated with the device, DBH tape and calliper were used to measure the DBH of the trees in the eight plots. The calculated results were used as reference values. Equations (9)–(14) were applied to calculate the DBH error (Error), bias (BIAS), relative bias (relBIAS), root-mean-square error (RMSE), relative root-mean-square error (relRMSE) and relative accuracy (relative accuracy) to estimate the availability of the measurement data, where $dbh_i$ is the data measured using the device and $DBH_i$ is the data measured using the DBH tape and calliper.

$$Error = dbh_i - DBH_i \tag{9}$$

$$BIAS = \frac{\sum_{i=1}^{n}(dbh_i - DBH_i)}{n} \tag{10}$$

$$relBIAS = \frac{\sum_{i=1}^{n}\left(\frac{dbh_i}{DBH_i} - 1\right)}{n} \times 100\% \tag{11}$$

$$RMSE = \sqrt{\frac{\sum_{i=1}^{n}(dbh_i - DBH_i)^2}{n}} \tag{12}$$

$$relRMSE = \sqrt{\frac{\sum_{i=1}^{n}\left(\frac{dbh_i}{DBH_i} - 1\right)^2}{n}} \times 100\% \tag{13}$$

$$relative\ Accuracy = 1 - relative\ RMSE \tag{14}$$

To evaluate the location estimation, the data measured using the compass were converted to derive reference values of the locations. First, the compass was accurately placed at the centre of the square-shaped sample plot, and the distance of each tree from the central point and angle from true north were measured to derive coordinate information via conversion. The BIAS and RMSE of the coordinates of the estimated points along the directions of the *X*- and *Y*-axes were calculated, and the straight-line range error (Ed) between the estimated and reference points was used to estimate the similarity:

$$Ed^2 = (x_i - X_i)^2 + (y_i - Y_i)^2 \tag{15}$$

where $x_i$ and $y_i$ are estimated values from the device, and $X_i$ and $Y_i$ are reference values from the compass measurement.

### 3.3. DBH Evaluation

The DBH values measured using the measurement instrument ($dbh_i$), traditional DBH tape ($DBH_{tape}$) and calliper ($DBH_{caliper}$) were compared for error analysis. The evaluation results are shown in Table 3. Compared with the data measured using the DBH tape, the overall bias of DBH measurement is −0.03 cm (−0.24%), RMSE is 0.13 cm (0.60%) and the overall measurement similarity is 99.40%. The linear regression plot in Figure 8a shows that the correlation coefficient ($R^2$) is 0.9981, and the distribution of bias in different diameter classes is shown in Figure 9a. Compared with data measured using the calliper, the overall BIAS is −0.29 cm (−1.43%), RMSE is 0.46 cm (1.75%) and the overall measurement similarity is 98.25%. The linear regression plot in Figure 8b shows that the $R^2$ is 0.9983, and the distribution of bias in different diameter classes is shown in Figure 9b. The evaluation results show that as DBH increases, bias increases, and the values measured with the device are consistent with those obtained using traditional tools.

**Table 3.** DBH evaluation.

| Plot No. | DHB Tape | | | | | Calliper | | | | |
|---|---|---|---|---|---|---|---|---|---|---|
| | BIAS (cm) | relBIAS (%) | RMSE (cm) | relRMSE (%) | relAcc (%) | BIAS (cm) | relBIAS (%) | RMSE (cm) | relRMSE (%) | RelAcc (%) |
| 1 | −0.02 | −0.06 | 0.25 | 1.29 | 98.71 | −0.20 | −1.10 | 0.34 | 1.79 | 98.21 |
| 2 | −0.01 | −0.12 | 0.06 | 0.70 | 99.30 | −0.13 | −0.68 | 0.29 | 1.73 | 98.27 |
| 3 | −0.02 | −0.12 | 0.05 | 0.35 | 99.65 | −0.47 | −2.63 | 0.50 | 2.77 | 97.23 |
| 4 | −0.05 | −0.07 | 0.06 | 0.28 | 99.72 | −0.35 | −1.44 | 0.55 | 2.38 | 97.62 |
| 5 | −0.07 | −0.39 | 0.15 | 0.75 | 99.25 | −0.45 | −1.83 | 0.63 | 2.88 | 97.12 |
| 6 | −0.03 | −0.16 | 0.06 | 0.36 | 99.64 | −0.20 | −0.82 | 0.45 | 1.79 | 98.21 |
| 7 | −0.10 | −0.48 | 0.16 | 1.43 | 98.57 | −0.34 | −2.15 | 0.63 | 4.61 | 95.39 |
| 8 | −0.07 | −0.47 | 0.11 | 0.88 | 99.12 | −0.33 | −1.86 | 0.41 | 2.22 | 97.78 |
| Total | −0.03 | −0.24 | 0.13 | 0.60 | 99.40 | −0.29 | −1.43 | 0.46 | 1.75 | 98.25 |

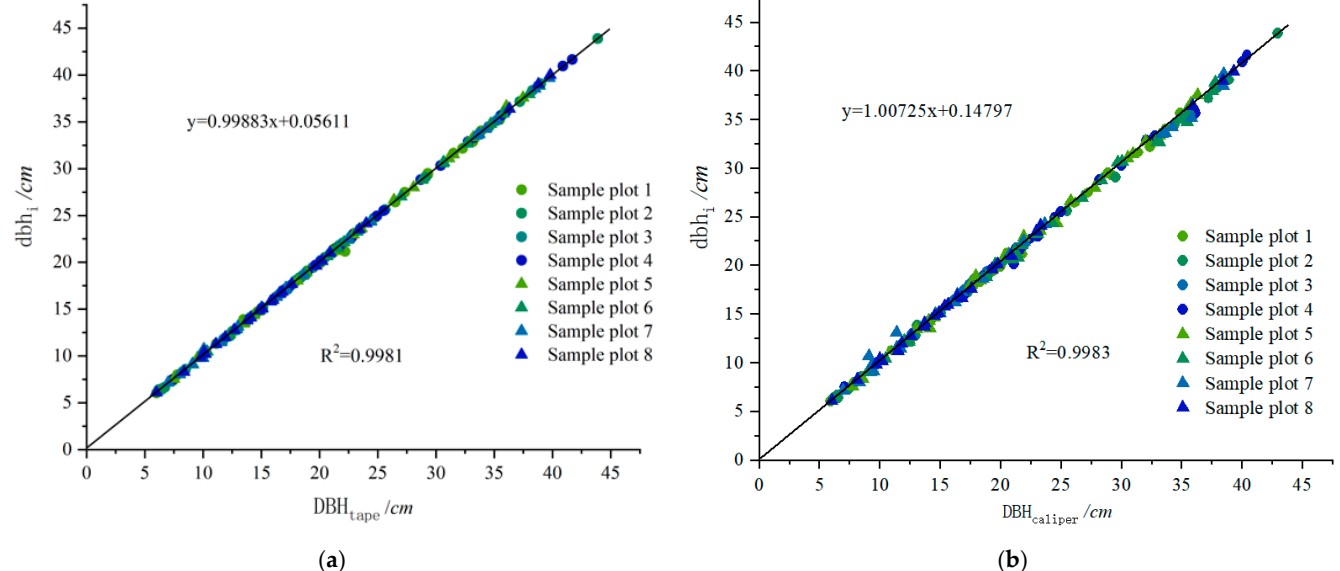

(**a**)      (**b**)

**Figure 8.** Linear fitting graphs of $dbh_i$ with (**a**) $DBH_{tape}$ and (**b**) $DBH_{caliper}$.

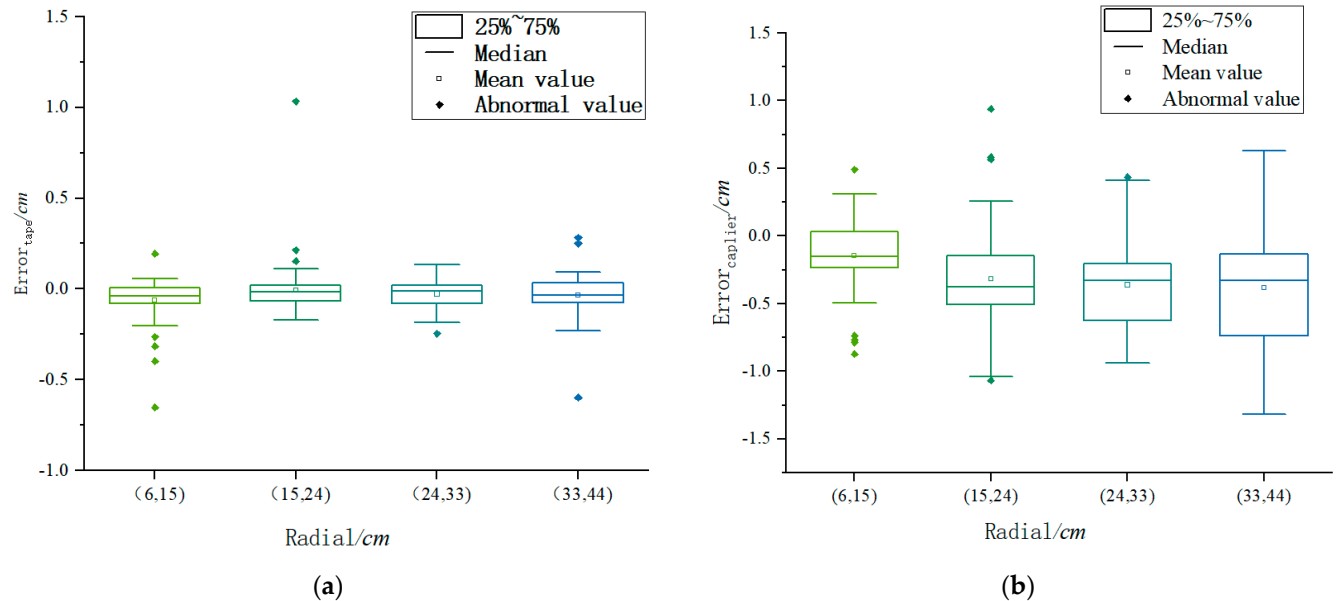

(**a**)      (**b**)

**Figure 9.** Error distribution of diameter with (**a**) DBH tape and (**b**) calliper.

### 3.4. Location Evaluation

The coordinates measured with the positioning device ($x_i$, $y_i$) were compared with those obtained using a compass ($X_i$, $Y_i$) for bias analysis. The distribution of bias of the distance values on the XOY plane (ED) are shown in Figure 10, and the evaluation of the distance biases (ED) is shown in Table 4. The evaluation results of the accuracy of the X- and Y-axes are shown in Table 5. It is shown that the ED range is 0.35 to 78.46 cm, the mean distance bias error is 25.41 cm and the standard deviation (Std) is 23.11 cm. On the X-axis, the bias range is −2.48 to 9.41 cm, the overall BIAS is 2.40 cm, the RMSE range is 8.83 to 27.90 cm, the overall RMSE is 23.22 cm and the overall measurement similarity is 93.95%. On the Y-axis, the BIAS range is −3.85 to 8.05 cm, the overall BIAS is 1.99 cm, the RMSE range is 17.18 to 33.08 cm, the overall RMSE is 25.77 cm and the overall measurement similarity is 92.55%. The slope of plots 1–5 is less than 10°, that of plots 6–8 is higher than 10° and their overall average ED bias is 30.55 and 31.79 cm, respectively. The overall EDs are similar, indicating the minimal effect of slope on the device positioning. Thus, the device can accurately position trees in different environments.

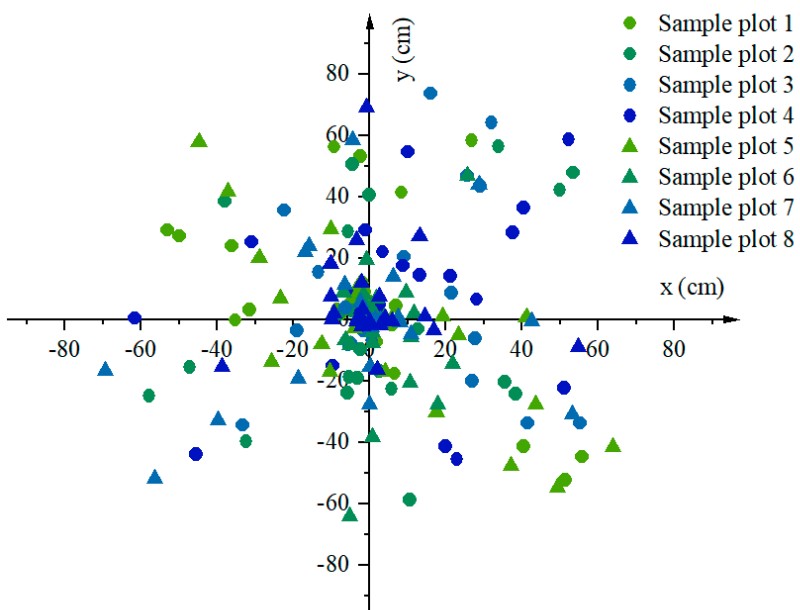

**Figure 10.** Range error (ED) distribution of bias.

**Table 4.** Range error (ED) evaluation of bias.

| Plot No. | ED (cm) | | | |
| --- | --- | --- | --- | --- |
| | Mean (cm) | Max (cm) | Min (cm) | Std (cm) |
| 1 | 28.14 | 73.43 | 1.70 | 25.21 |
| 2 | 30.12 | 71.68 | 0.87 | 23.84 |
| 3 | 33.97 | 75.36 | 3.97 | 23.15 |
| 4 | 32.35 | 78.46 | 2.20 | 23.04 |
| 5 | 28.74 | 76.23 | 1.03 | 24.23 |
| 6 | 13.95 | 64.22 | 0.35 | 16.33 |
| 7 | 23.77 | 76.66 | 0.43 | 24.33 |
| 8 | 15.07 | 68.97 | 0.44 | 17.55 |
| Total | 25.41 | 78.46 | 0.35 | 23.11 |

**Table 5.** Evaluation of accuracy of *X*- and *Y*-axes.

| Plot No. | X (cm) | | | Y (cm) | | |
|---|---|---|---|---|---|---|
| | BIAS (cm) | RMSE (cm) | RelAcc (%) | BIAS (cm) | RMSE (cm) | RelAcc (%) |
| 1 | −1.52 | 25.40 | 93.24 | 6.56 | 27.54 | 94.65 |
| 2 | 1.07 | 25.25 | 92.34 | 0.35 | 28.60 | 85.07 |
| 3 | 9.41 | 24.99 | 91.29 | 8.05 | 33.08 | 93.71 |
| 4 | 6.80 | 27.90 | 94.57 | 6.81 | 27.85 | 95.23 |
| 5 | 4.50 | 27.32 | 94.64 | −3.85 | 25.36 | 92.40 |
| 6 | 3.25 | 8.83 | 96.21 | −3.41 | 19.33 | 97.07 |
| 7 | −2.48 | 25.04 | 93.75 | −1.14 | 22.52 | 94.35 |
| 8 | 0.81 | 15.09 | 97.32 | 4.80 | 17.18 | 96.55 |
| Total | 2.40 | 23.22 | 93.95 | 1.99 | 25.77 | 92.55 |

*3.5. Measurement Efficiency Evaluation*

The DBH, location and time measured with the device were compared with those measured using the tape, calliper and range compass. The evaluation of measurement efficiency is shown in Table 6. The actual measurement was conducted in a certain sequence. The traditional measurement involved three personnel, two for measuring DBH with DBH tape and calliper and one for recording the time. Subsequently, they collaboratively measured tree locations using the calliper and recorded the time. Finally, the recorded data were entered into a computer, and the time was recorded. The integrated measurement device was operated by one person to measure DBH, location and other data, then the data were uploaded and the time was recorded. The average time per tree measurement with the integrated measurement device was 18.53 s, that using the DBH tape and ranging compass was 178.68 s and that using the calliper and ranging compass was 181.55 s. Hence, the efficiency of the integrated measurement device was 8.64 times higher than that of the DBH tape and ranging compass and 8.79 times higher than that of the compass and ranging compass.

**Table 6.** Measurement efficiency evaluation.

| Measurement Tool | Measuring Time per Tree (s) | Total Time (s) |
|---|---|---|
| Integrated device | 18.53 | 3706 |
| Tape + ranging compass | 178.68 | 35,736 |
| Calliper + ranging compass | 181.55 | 36,310 |

## 4. Discussion

Forest surveys have evolved with advances in sensor technology, microelectronics, remote sensing, machine learning, computer vision and Internet of Things technologies. In recent years, different devices and methods for measuring DBH have emerged. Fan et al. used the LiDAR method to estimate tree DBH, and the experimental data showed a bias of 0.38 cm and 2.75%, and an RMSE of 1.28 cm and 9.28% [5]. This approach is mainly applied to single trees; the data processing method is complicated, the equipment is costly, users need professional training and it cannot be widely generalised for production practice. Hyyppa et al. used Kinect-derived tree diameter measurement, which agreed with DBH tape measurements, with an RMSE of 1.9 cm and 7.3% [8]. However, the authors stated that this method is only suitable for single trees and requires professional manipulation. Sun et al. made a DBH measuring device using a draw-wire sensor, which had an accuracy of 99.97% [10]. The device in this paper is only a preliminary model and has not yet been properly packaged, designed and tested. Oveland et al. used a moving terrestrial laser scanner to automatically estimate tree position and stem diameter. The accuracy of their equipment showed an RMSE of 1.5 cm and 7.5%. The authors noted that the equipment is

costly and complex to operate, and the recognition efficiency is significantly reduced for thicker woods and smaller trees [17].

Various researchers are using different approaches to improve the methods and efficiency of measuring tree DBH, and they have achieved certain results. However, there are some problematic issues that cannot be resolved, such as the cost, lack of advancement or difficult operation of the equipment, and tree DBH or location measurement is a one-sided factor. A few years ago, our team introduced a fast DBH tree-height measurement device. Compared with the DBH tape, the DBH measurement has a bias of $-0.03$ cm and a root-mean-square error (RMSE) of 0.69 cm for this device. Compared with the calliper, the DBH measurements have a 0.16 cm bias and a 0.46 cm RMSE. The DBH measurement has a bias of 0.16 cm and an RMSE of 0.46 cm [13]. The article mentions that the device has major limitations. One is that it can only be used in relatively empty environments, and the other is that it is severely affected by environmental conditions, with the device not working properly in relatively strong light or relatively hot or cold environments. Our team developed a new device last year, and we compared its measurement results with those of the DBH tape and calliper. Their RMSE values were 0.33 cm (1.57%) and 0.52 cm (2.65%), respectively [14]. The device needs to replace the contact when measuring trees with a diameter of 15.6 cm or more; it is easy to encounter large errors when measuring trees with a diameter of 60.6 cm or more, and the device can only perform diameter measurements. As for positioning, compared with the previous equipment, the device adopts a newly developed algorithm, does not need to be placed in a fixed position in the sample field and its measurement range has changed from 10 m × 10 m to 24 m × 24 m without any reduction in the accuracy of the measurement data. Based on previous research [21], we continued to improve the equipment, innovating the mechanical structure and diameter calculation methods of the original device to improve the efficiency and stability and developing equipment to conduct the most difficult and time-consuming tree-position measurement functions in forest surveys through repeated experiments. Then, we conducted field research to test the use of the equipment. Actual measurements were conducted on eight 24 m × 24 m sample plots with different slope values. Compared with the traditional DBH tape measurements, the device had an overall bias of $-0.24\%$ and a measurement similarity of 99.6%. Compared with the traditional calliper, the device had an overall bias of $-1.43\%$ and a measurement similarity of 98.25%. Compared with the compass, the device had an overall bias of $-1.32\%$ and a measurement similarity of 93.55% along the *X*-axis and an overall bias of $-1.03\%$ and a measurement similarity of 92.55% along the *Y*-axis. In addition, the ranging error (ED) was 0.35–78 cm, and the mean ED was 25.41 cm. The measuring devices from Haglöf Company meet the requirements for DBH and position measurements. The error of position-measuring devices is 15–30 cm, and the price is about CHF 8328.4, so they are difficult to popularise and widely use [20].

In this study, we designed an entirely new tree diameter and position measurement system combining angle sensors and UWB modules, along with the upper computer software. The data from the device can be directly imported into the computer to form tables and pictures with one click, and the software has primary data analysis capability. The equipment as a whole can be tucked into a backpack and carried. The device has a length of 63.5 cm, a width of 17.7 cm, a height of 20.3 cm and a weight of 3.82 kg. The cost of the device is approximately CHF 212.5, which is in line with the standards of forestry measuring equipment. Relative to devices that require manual measurements, it is convenient to carry and avoids tedious manual data entry while ensuring accuracy. Practical use has demonstrated that the device has preliminary waterproof capacity and the data-measuring device operates naturally in rainy weather. In temperature simulation experiments, the equipment has excellent performance in the range of $-17$ to 50 °C, and it can be used continuously for more than 20 h on a single charge without failure. After inspection, the measurement error of the equipment in the diameter of the standard circle was 0.001 cm. The actual measurement proved that, compared with the traditional manual measurement, the device had improved efficiency by at least eight times and can effectively

save survey and measurement time and cost, and the equipment can host computer settings to avoid errors in uploading data twice, and has a certain promotional and practical value.

## 5. Conclusions

We designed a device that combines different sensors using a novel algorithm and verified its reliability. It can achieve integrated measurement of tree factors such as DBH and position. After field measurements of 200 trees in different stands on eight 24 m × 24 m samples, it was found that the measurements of this device were similar to those of the DBH tape and calliper, and the efficiency was more than eight times better. The device has the advantages of easy portability, simple operation and high efficiency. The measuring equipment is portable and practical for forest surveys, and combined with the use of the computer, data can be uploaded and analysed with one click. However, the device still has some shortcomings and needs to be improved, and it has not been tested in a sample plot with a slope greater than 20° or in a more complex forest environment.

**Author Contributions:** Conceptualisation, L.F.; Formal analysis, S.L.; Resources, L.F., X.C. and C.L.; Validation, Y.S., S.L. and F.Y.; Investigation, F.Y.; Writing—original draft, Y.S.; Writing—review and editing, S.L. All authors have read and agreed to the published version of the manuscript.

**Funding:** This research was funded by the Zhejiang Provincial Key Science and Technology Project (2018C02013), the National Natural Science Foundation of China (No. 42001354) and the Scientific Research Project of the Education Department of Zhejiang Province (Y202250137).

**Data Availability Statement:** The data used to support the findings of this study are available from the corresponding author upon request.

**Conflicts of Interest:** The authors declare no conflict of interest.

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
