# Peer review of "Method and Device for Measuring the Diameter at Breast Height and Location of Trees in Sample Plots"

_forests, doi:10.3390/f14091723_

Round 1

Reviewer 1 Report (New Reviewer)

I received the following document sent to Forests: Method and device for a sample plot factor for the diameter at breast height and location of trees (forests-2546046). After analyzing it, everything indicates that it is a working version, not yet fully prepared by the Authors' team for publication. This is confirmed by the already inappropriate title in the context of the content of the document. In the Introduction and in the Discussion there are fragments of the text highlighted, prepared by the Authors for correction, their analysis by the reviewer confirms this necessity. The bibliography is anonymous, only the initials of the authors are given, it is not according to the MDPI standard. Upon investigation, I find that an earlier publication is missing from the list:

Linhao Sun, Luming Fang, Yuhui Weng, Siqing Zheng. An Integrated Method for Coding Trees, Measuring Tree Diameter, and Estimating Tree Positions. Sensors 2020, 20, 144; doi:10.3390/s20010144

from which there are many references in this document, for example in Chapter 2. Material and Methods: Table 1, Figure 2, 6 and 7 etc.

Extensive editing of English language required.

Best regards,

Reviewer

Extensive editing of English language required.

Author Response

Dear Reviewer,

Thank you very much for your suggestions and comments on our manuscript. With the corresponding modifications, these suggestions are of great help to the improvement of our manuscript. We have studied comments carefully and have made correction which we hope meet with approval. The main corrections and responses in the paper are as flowing:

Point 1: The bibliography is anonymous, only the initials of the authors are given, it is not according to the MDPI standard.

Response: ​I am truly sorry for the mischaracterization of the references and for the inconvenience caused by your examination of the manuscript. I may have misunderstood the presentation in the template and revised it all after referring to the others paper.

Point 2: Upon investigation, I find that an earlier publication is missing from the list:

Linhao Sun, Luming Fang, Yuhui Weng, Siqing Zheng. An Integrated Method for Coding Trees, Measuring Tree Diameter, and Estimating Tree Positions. Sensors 2020, 20, 144; doi:10.3390/s20010144

from which there are many references in this document, for example in Chapter 2. Material and Methods: Table 1, Figure 2, 6 and 7 etc.

Response: Thanks to the reminder, the article has now been added to the literature, and the reference is cited in several places. The content of each table and picture is different from the previous one, only the previous expression is referenced, and comparisons are added in the discussion section (lines 284-292). This is the first time that we have tried to combine multiple trees factors many years ago. The overall size of the equipment is large, with steep limitations on usage. Now we have overcome some of these difficulties. First, the size of the device was adjusted to make it easier to carry. Second, a different algorithm is used to ensure accuracy and significantly increase the measurement range. Third, the use of the device has also become simple, and data measurements can be started at any location in the sample.​

Point 3: Extensive editing of English language required.

Response: Thank you for the suggestion. My English expressions may not be great enough. Now that I have applied to the official English editing function of MDPI to revise the article, I have now uploaded my revised manuscript and marked some critical revisions in the article.

With best regards,

Yours Sincerely,

Shangyang Li

Reviewer 2 Report (New Reviewer)

General comments:

1.      The main contribution of the manuscript is to propose a device that integrates today's modern technologies to rapidly measure DBH size and tree height.

2.      The topic is very interesting because it significantly speeds up the measurements of tree sizes and positions and their transfer to the computer.

3.      Unfortunately, the authors falsely claim to present a statistical method for evaluating the accuracy of a new device. Authors should be aware that they are only comparing measurements obtained with a newly developed device and conventional methods of measurement. For accuracy assessment, authors must provide exact values ​​of tree sizes. Therefore, the discussion that the measurements provided by the new device are more accurate has no statistical confirmation.

4.      It should be noted that the authors do not have a control (accurate measurements) database for accuracy assessment.

5.      In view of points 3 and 4, chapters 3, 4, and 5 should discuss the comparison of the two measurement methods.

Specific comments:

Page 5: Supplement the literature on the derivation of equations 1 and 2.

Page 8: ... where dbhi is the data measured using the device and dbhi is the

data measured using the DBH tape and caliper... Repetitive, fix it.

The article is written in acceptable language.

Author Response

Dear Reviewer,

Thank you very much for your suggestions and comments on our manuscript. With the corresponding modifications, these suggestions are of great help to the improvement of our manuscript. We have studied comments carefully and have made correction which we hope meet with approval. The main corrections and responses in the paper are as flowing:

Point 1: Unfortunately, the authors falsely claim to present a statistical method for evaluating the accuracy of a new device. Authors should be aware that they are only comparing measurements obtained with a newly developed device and conventional methods of measurement. For accuracy assessment, authors must provide exact values ​​of tree sizes. Therefore, the discussion that the measurements provided by the new device are more accurate has no statistical confirmation.

Response: On Forestry, tree DBH measurement generally consider the value measured by the tape as the standard value of the DBH of the tree and the value measured by the caliper as the reference value when measuring the DBH. The main purpose of our research is to replace the tedious manual work with better electronic equipment, so the measured values of our equipment are compared with those measured by tape and caliper to verify the effectiveness of the equipment. Perhaps our statement is not accurate enough, so in the article 200-203 lines, we specifically indicate that the tape and calipers are used as standard and reference values, and then we compare our equipmen measured values with them.

Point 2: It should be noted that the authors do not have a control (accurate measurements) database for accuracy assessment.

Response: According to forestry reality, We determined that the data measured by the tape measure were standard values, so we selected a total of about 300 trees at eight different sample points for data collecting testing. The amount of data may be slightly less, which is enough to prove the effectiveness of the device and reflect the problem from the experimental results. We will continue to advance research to find a better way to address this issue. Finally, we also tested our equipment. If it is a standard circle, our accuracy can reach 0.001 cm. In the article, line 314, this is the exact value measurement we can do.

Point 3: In view of points 3 and 4, chapters 3, 4, and 5 should discuss the comparison of the two measurement methods.

Response: ​ In the third chapter, we made a comparative analysis of the data. Mainly, we believed that the values of the tape measure and caliper were accurate, so we carried out error calculation, deviation range calculation and root-mean-square error calculation between the equipment and the manual measurement means, and drew pictures and tables. From the pictures and tables, we could conclude that the accuracy of our equipment was within the controllable range. Meet the use of forest surveys.

Point 3: Page 5: Supplement the literature on the derivation of equations 1 and 2.

Response: ​Thank you so much for pointing this out.  On line 122 of the paper, we have added the original reference in the appropriate place to explain the source and derivation of the formula.​

Point 4: Page 8: ... where dbhi is the data measured using the device and dbhi is the

data measured using the DBH tape and caliper... Repetitive, fix it.

Response: I sincerely apologize for my oversight, which has now been corrected on page 8, line 185.

With best regards,

Yours Sincerely,

Shangyang Li

Round 2

Reviewer 1 Report (New Reviewer)

General remarks of the reviewer

The revised version of the manuscript presented by the Authors has already the character of a scientific article. Its subject matter is important for the practice of forest measurement. Each technological innovation in this area is of utilitarian importance in order to reduce labor intensity while maintaining satisfactory accuracy.

The following chapters requires some clarification:

In the Introduction, there are practically no reports from America and Europe, which later has consequences in a limited Discussion in the context of the results obtained by the Authors.

The Discussion pointed to the effectiveness of the developed dbh measurement and tree positioning technology in the context of the simplest manual method: an ordinary caliper and a compass. However, in the case of terrestrial measurements, e.g. in Europe, as part of a full assessment on circular plotss, electronic calipers and sets of transponders mounted on telescopes are used to determine dbh, height and position of trees. And in such a combination it would be interesting to compare the efficiency of the technology.

In the Conclusion, it is necessary to comment for practice that the developed technology is based on relatively small empirical material. Based on the analysis of Table 2: 200 trees in the dbh range of 6-44 cm with a low density of 1 tree per 30 m2 (Plot 3), - 21 m2 (Plot 2), on average 1 tree per 23 m2.

Technical Notes

Throughout the text, use the term "caliper" consistently and not, for example, on lines 33-34 "girth tape".

In Table 2, give the name of the pine species (S6).

In Table 4, explain the shortcut "Sad", or did you mean "Sd"?

In Figure 9 on the X axis mark the traditional dbh ranges (6-15, 15-24 etc.).

Make all Figures  clearer if possible.

Eliminate logical errors and typos.

The description of the literature item needs to be corrected as required by the publisher: articles, books and other sources - italics of journal titles, year in bold, correct pages of journals and the access link and date of access in English. According to MDPI standard.

Details in the attached manuscript.

Moderate editing of English language required.

Author Response

Dear Reviewer,

​Many thanks for your comments on the previous manuscript. I hope the manuscript will be approved by you, and I wish to express here my most sincere blessing.

Point 1: In the Introduction, there are practically no reports from America and Europe, which later has consequences in a limited Discussion in the context of the results obtained by the Authors.

The Discussion pointed to the effectiveness of the developed dbh measurement and tree positioning technology in the context of the simplest manual method: an ordinary caliper and a compass. However, in the case of terrestrial measurements, e.g. in Europe, as part of a full assessment on circular plotss, electronic calipers and sets of transponders mounted on telescopes are used to determine dbh, height and position of trees. And in such a combination it would be interesting to compare the efficiency of the technology.

Point 2: In the Conclusion, it is necessary to comment for practice that the developed technology is based on relatively small empirical material. Based on the analysis of Table 2: 200 trees in the dbh range of 6-44 cm with a low density of 1 tree per 30 m2 (Plot 3), - 21 m2 (Plot 2), on average 1 tree per 23 m2.

Response: In the conclusion ( lines 329-330), a description of the size of the plot was added, with a total of 200 trees in eight plots of different stands of 24 m × 24 m.

Point 3: Throughout the text, use the term "caliper" consistently and not, for example, on lines 33-34 "girth tape".

Response:Thanks for your reminder, a correction has been made on lines 33-34 of this article.

Point 4: In Table 2, give the name of the pine species (S6).

Response: Thanks for your reminder, the correction has been made below Table 2 of the article.

Point 5: In Table 4, explain the shortcut "Sad", or did you mean "Sd"?

Response: sad refers to the standard deviation, An explanation of this abbreviation has now been added in line 224 of the article.

Point 6: In Figure 9 on the X axis mark the traditional dbh ranges (6-15, 15-24 etc.).

Make all Figures  clearer if possible

Response: Figure 9 is the level diagram of the error box. Our equipment and traditional equipment use the same X-axis and display errors in different ranges, which may be because the picture is not clear enough. We reloaded each image, resized and resized the image so that the numbers on each image became clearer.

Point 7:Eliminate logical errors and typos.

The description of the literature item needs to be corrected as required by the publisher: articles, books and other sources - italics of journal titles, year in bold, correct pages of journals and the access link and date of access in English. According to MDPI standard.

Response: We reread the article and made corrections to the inconsistencies, in particular We have changed the format of all the References, the journal names are used in italics, the year of publication is used in bold, and then the access link is added.

Reviewer 2 Report (New Reviewer)

General comments:

1.      Unfortunately, the authors did not pay proper attention to the reviewer's comments.

2.      …The results show that the accuracy of the device and the traditional tape and caliper was 99.26% and

20 98.25%, respectively…. Statistically, the statement in the sentence is meaningless. The meaning of the word 'accuracy' is not defined in the article. You are probably talking about the coefficient of determination, which shows what part of the 'caliper' or 'tape' measurements is covered by the measurements obtained with the new device. Since you claim that 'caliper' and 'tape' measurements are accurate - control, you cannot claim that measurements with the new device are more accurate than 'caliper' or 'type'.

3.      Equation 14 defining the new concept 'relative accuracy' will almost always be negative, what is the point of subtracting the quantity expressed as a percentage from 1.

4.      In Tables 3 and 5, it is necessary to indicate whether the 'bias' is significant or not.

The article is written in acceptable language.

Author Response

Dear Reviewer,

Thank you very much for your suggestions on our manuscript. The main corrections and responses in the paper are as flowing:

Point 1: …The results show that the accuracy of the device and the traditional tape and caliper was 99.26% and20 98.25%, respectively…. Statistically, the statement in the sentence is meaningless. The meaning of the word 'accuracy' is not defined in the article. You are probably talking about the coefficient of determination, which shows what part of the 'caliper' or 'tape' measurements is covered by the measurements obtained with the new device. Since you claim that 'caliper' and 'tape' measurements are accurate - control, you cannot claim that measurements with the new device are more accurate than 'caliper' or 'type'.

Response: After discussion by our team, we decide to use 'similarity '. The initial goal of our design equipment is to conduct the survey of forestry resources more efficiently. Why did we say he was more accurate in the first place, because after manual measurement of data, a lot of data is likely to have human errors, resulting in data errors. But for the data measured by the device, we now choose 'similarity ', which may have a higher degree of rigor. We have revised the description of the introduction and conclusion to make the sentences more rigorous.

Point 2: Equation 14 defining the new concept 'relative accuracy' will almost always be negative, what is the point of subtracting the quantity expressed as a percentage from 1.

Response: Thanks a lot for your kind suggestion. We compared all the data in different ways, compared the difference between the measured value of the equipment and the traditional equipment, and observed whether it can replace the traditional equipment. The relative accuracy is listed in columns 6 and 11 of the table, indicating the similarity of the measured value of the equipment with the traditional equipment. Table three, columns two and three, and columns seven and eight negative number is ‘relBIAS’, show that the diameter measurement value of our equipment is slightly smaller than that of traditional equipment.

Point 2: In Tables 3 and 5, it is necessary to indicate whether the 'bias' is significant or not.

Response: Now we have shown the deviation values of Table 3 and Table 5 in bold, we have drawn the box drawing of the deviation in Figure 9, and the deviation shown in the pictures and tables is within the acceptable range.

This manuscript is a resubmission of an earlier submission. The following is a list of the peer review reports and author responses from that submission.

Round 1

Reviewer 1 Report

In the introduction, the studies on terrestrial lidar should be introduced and cited in order to stimulate discussion of the position of the trees and the measurement of diameter at breast height.

The discussion section is very poor and does not allow the quality of the work to be appreciated.

The author is right to mention the measurement time. He should also allude to the overall cost of obtaining this equipment and the software licences used.

Author Response

Dear Reviewer,

Thank you very much for your suggestions and comments on our manuscript. With the corresponding modifications, these suggestions are of great help to the improvement of our manuscript. We have studied comments carefully and have made correction which we hope meet with approval. The main corrections and responses in the paper are as flowing:

Point 1: The discussion section is very poor

Response: Thank you so much for your advice. We have rewritten the discussion section in which the advantages and disadvantages of various tree diameter measurements and their accuracy are compared. In addition, we have detailed the parameters and advantages of the device in the hope of gaining your approval.

Point 2: He should also allude to the overall cost of obtaining this equipment and the software licences used.

Response: We are extremely sorry for our oversight of this matter. In the discussion section of the article, inspired by your comments, various parameters of the device have been added, including the price. Our devices and system interfaces were developed and written by ourselves. Software copyrights are being applied for. Thank you for your comments.

Reviewer 2 Report

General comments

The article does not follow the usual structure of a scientific article: introduction, material and methods, results, discussion and conclusions.

It looks more like an instruction manual for the device and the scientific interest appears only at the end of the article (page 9) and is very low. The results are very messy and the discussion insufficient. The conclusions are also very brief.

I recommend rejecting its publication in Forests.

Author Response

Dear Reviewer,

Thank you very much for your suggestions and comments on our manuscript. With the corresponding modifications, these suggestions are of great help to the improvement of our manuscript. We have studied comments carefully and have made correction which we hope meet with approval. The main corrections and responses in the paper are as flowing:

Point 1: The article does not follow the usual structure of a scientific article: introduction, material and methods, results, discussion and conclusions.

Response: We are extremely sorry for their negligence. We have revised the paper, removing parts and rewriting the conclusion and discussion sections, adding some necessary parameter comments.

Point 2: It looks more like an instruction manual for the device and the scientific interest appears only at the end of the article (page 9) and is very low.

Response: ​ ​Our device detects two different forest factors, tree DBH and tree position, and the algorithms and experimental methods used are different, so there will be a large amount of data and graphs to demonstrate the reliability of the device and the accuracy of the device algorithm. We now modify some of the constructions and diagrams to make the paper clearer.

Point 3: The results are very messy and the discussion insufficient. The conclusions are also very brief.

Response: ​Many thanks for your valuable comments. We have rewritten the discussion section in which the advantages and disadvantages of various tree diameter measurements and their accuracy are compared. In addition, we have detailed the parameters and advantages of the device in the hope of gaining your approval.

Round 2

Reviewer 2 Report

The authors have made great improvements to the article. They have removed many paragraphs that looked like instructions for the device and have greatly improved the discussion and conclusions.

 However, the article still does not have the correct structure of a scientific article: Introduction, Material and Methods, Results, Discussion and Conclusions.

There are some sections such as Device design and Experimental analysis that should be included Material and Methods. The Results chapter is missing.

The authors must correct these errors so that the article can be published.

Author Response

Dear Reviewer,

Thank you very much for your suggestions and comments on our manuscript. With the corresponding modifications, these suggestions are of great help to the improvement of our manuscript. We have studied comments carefully and have made correction which we hope meet with approval. The main corrections and responses in the paper are as flowing:

Point 1: However, the article still does not have the correct structure of a scientific article: Introduction, Material and Methods, Results, Discussion and Conclusions.

Response: We are extremely sorry for their negligence. We have made changes to the format of the article, in the order of Introduction, Material and Methods, Results, Discussion and Conclusions.

Point 2: There are some sections such as Device design and Experimental analysis that should be included Material and Methods. The Results chapter is missing.

Response: ​ ​Now we will start with the introduction, and then show the mechanical structure of the equipment, the algorithm of tree diameter measurement and the algorithm of tree position in the Material and Methods section. Then we will show the process of equipment experiment, and then the test results of the equipment, and then discuss and Conclusions. I hope to get your approval
